# High Intensity Focused Ultrasound Ablation for Juvenile Cystic Adenomyosis: Two Case Reports and Literature Review

**DOI:** 10.3390/diagnostics13091608

**Published:** 2023-05-01

**Authors:** Xin Liu, Jingxi Wang, Yanglu Liu, Shuang Luo, Gaowu Yan, Huaqi Yang, Lili Wan, Guohua Huang

**Affiliations:** 1School of Medical and Life Science, Chengdu University of Traditional Chinese Medicine, Chengdu 610000, China; 2Department of Gynecology, Suining Central Hospital, Suining 629000, Chinayanghuaqi751@outlook.com (H.Y.);; 3Department of Radiology and Imaging, Suining Central Hospital, Suining 629000, China

**Keywords:** juvenile cystic adenomyosis, high-intensity focused ultrasound (HIFU), adolescent, dysmenorrhea

## Abstract

Cystic adenomyosis is a rare type of uterine adenomyosis, mainly seen in young women, which is often characterized by severe dysmenorrhea. The quality of life and reproductive function of young women could be affected by misdiagnosis and delayed treatment. At present, there are no universal guidelines and consensus. We report two cases of patients with cystic adenomyosis in juveniles treated with high-intensity focused ultrasound (HIFU) ablation. In the first case, magnetic resonance imaging (MRI) indicated a cystic mass of 2.0 cm × 3.1 cm × 2.4 cm in the uterus. After she underwent HIFU treatment, her pelvic MRI showed a mass of 1.1 × 2.4 cm in size, and her dysmenorrhea symptoms gradually disappeared. In the second case, a pelvic MRI indicated a 5.1 cm × 3.3 cm × 4.7 cm cystic mass in the uterus. After she underwent HIFU and combined four consecutive cycles of GnRH-a treatment, the lesion shrunk 1.2 cm ×1.4 cm × 1.6 cm, without dysmenorrhea. Simultaneously, the report reviewed 14 cases of juvenile cystic adenomyosis over the last ten years. HIFU or HIFU-combined drugs were safe and effective in treating juvenile cystic adenomyosis, but multicenter and prospective studies may be necessary to validate this in the future.

## 1. Introduction

Adenomyosis is a common gynecologic disease that mainly occurs in women aged thirty to fifty years and is characterized by the invasion of ectopic endometrial tissue into the myometrium. Myometrial cysts of any size are the direct ultrasonographic features of adenomyosis, and transvaginal 2D ultrasound is the most accurate way to observe this sign [1,2]. However, the lesions of the rare cystic adenomyosis have a diameter of more than 1 cm and are filled with ectopic endometrial tissue and bloody fluid [3]. According to literature reports, the disease has increased in adolescent patients in recent years. We reviewed 14 cases of adolescent cystic adenomyosis in the past decade (Table A1) and found that there was no unified expert consensus on the treatment of adolescent cystic adenomyosis, and the choice of treatment was difficult. Currently, the effect of medication is limited in patients with juvenile cystic adenomyosis. Surgical treatment is a mainly conservative operation with reproductive function preservation, which causes side effects such as adhesion, iatrogenic endometriosis, and a high recurrence rate [4]. Because juvenile cystic adenomyosis patients have fertility requirements, minimally invasive or noninvasive treatments are acceptable. Noninvasive treatment has been of gradually increased in the treatment of cystic adenomyosis. In this report, our center carried out two cases of juvenile cystic adenomyosis treatments using HIFU and HIFU combined with medication.

### 1.1. Case 1

The patient was 16 years old, without a sexual life history, and was admitted to the hospital with dysmenorrhea for 1 year, uterine occupation for half a year, and abdominal pain for 4 days. The patient had regular menstruation with severe dysmenorrhea. Four days earlier, she developed severe paroxysmal abdominal pain without an obvious trigger, and the VAS score was 7, without any other discomfort (Table 1). She denied having a history of hypertension, diabetes, heart disease, malignancy, and surgery. A physical examination revealed an anterior uterus of normal size and medium texture that was movable and painful pressure. Routine blood tests were normal, and serum tumor markers were CA125: 127.6 U/mL, CA199: 14.5 U/mL, and CEA: 0.3 ng/mL. A transabdominal ultrasound (Philips IU22, Philips) revealed an abnormal uterine development and a bicorned uterus. A posterior pelvic MRI showed a round cystic mass with a size of about 2.0 cm × 3.1 cm × 2.4 cm between the muscle walls of the right lateral wall of the uterus, showing short T1 and long T2 signals. The lesion was significantly enhanced on enhanced scanning, with clear boundaries and uterine cavity compression, which was considered to be cystic adenomyosis (Figure 1). Combining the patient’s clinical symptoms and MRI, the patient was diagnosed with juvenile cystic adenomyosis. Because of a strong desire for conservative treatment, she opted for HIFU treatment on 22 November 2017. A focused ultrasonic ablation was performed under sedation and analgesia, and compound polyethylene glycol electrolyte powder was used to induce diarrhea 3 days before treatment. Before the operation, routine skin preparation, degreasing, degassing, catheterization, and indwelling catheter were performed to fill the bladder and establish a safe acoustic channel. The patient was placed in a prone position and given fentanyl citrate and midazolam for sedation and analgesia. The location of the lesion was determined using ultrasound, and the therapeutic power was 200 W. The range of gray change in the lesion was satisfactory at 250 s after irradiation. The ultrasound showed no obvious blood flow signal in the lesion, and ultrasound angiography showed no blood perfusion in the lesion (Figure 2). The details of the procedure were treatment time: 49 min, irradiation time: 250 s, treatment power: 200 W, treatment intensity: 306/s, and an ablation volume evaluation of 80% (Table 2). During the treatment, the patient complained of pain in the treatment area and sacrococcygeal pain, which was relieved after rest. After the operation, the patient’s vital signs were stable, and she was advised to rest in a prone position for 2 h and fast from food and drink for 2 h. For the follow-up: the first month after treatment, the pelvic MRI showed a mass of 1.2 cm × 1.9 cm × 1.7 cm (Figure 3), with a reduction rate of 73%. The patient’s menstrual volume was the same as before, and her dysmenorrhea symptoms were significantly relieved with a VAS score of 2. Since then, irregular reexamination with ultrasound indicated that the size of the mass was 3–5 cm. The fifth year after treatment, pelvic MRI indicated that the mass was 1.1 cm × 2.4 cm × 1.0 cm (Figure 4), and the volume reduction rate was 69%. However, her menstruation was normal, her dysmenorrhea disappeared, and her VAS score was 0 (Table 1).

### 1.2. Case 2

The patient was 20 years old, without a sexual life history. She had normal menstrual cycles with menometrorrhagia and severe dysmenorrhea. She was admitted to the hospital with dysmenorrhea with a progressive aggravation over 5 years. The patient developed dysmenorrhea with progressive aggravation 5 years previously without obvious triggers, and the VAS score was 8 (Table 1). She had been treated with ibuprofen without relief in symptoms. The patient denied a history of hypertension, diabetes, heart disease, malignant tumor, and surgery. A physical examination revealed an enlarged uterus with a medium texture and pressure pain. Routine blood tests and serum tumor markers were normal. A transabdominal ultrasound (Voluson E10, GE Healthcare, Zipf, Austria) revealed a cystic lesion measuring 1.7 cm × 0.9 cm × 1.8 cm in the myometrium. The results of the CDFI: punctate and obvious strip blood flow signals. However, MRI clearly revealed a cystic lesion measuring 5.1 cm × 3.3 cm × 4.7 cm in the myometrium. The enhanced scan indicated a stale haemorrhage in the cystic lesion (Figure 5). According to the patient’s history and an auxiliary examination, the patient was diagnosed with juvenile cystic adenomyosis. Because of a strong desire for conservative treatment, she opted for HIFU treatment on 8 April 2022, under sedation and analgesia. The treatment power was 300 W, the irradiation time was 750 s, the treatment intensity was 517/s, and the treatment time was 75 min (Table 2). After the treatment, an ultrasound revealed an overall grayscale change in the lesion, and ultrasound angiography indicated a satisfactory ablation. The ablation volume was evaluated at 90%. The patient’s vital signs were stable after treatment. For the follow-up: the pelvic MRI on the first day after treatment showed a uterine cyst of 3.8 cm × 2.6 cm × 3.9 cm (Figure 6), with a reduction rate of 51% in mass volume. One month later, the patient had a menstrual flow of the same volume as before, with relief of dysmenorrhea symptoms and a VAS score of 4. In the second month after treatment, GnRH-a therapy was started for four cycles (28 days for one cycle). Eight months after treatment, a pelvic MRI showed a cystic mass of about 1.2 cm × 1.4 cm × 1.6 cm in size (Figure 7), and the volume reduction rate was 97%. Meanwhile, the patient had normal menstruation without dysmenorrhea, with a VAS score of 0 (Table 1).

## 2. Discussion

In 1990, Parulekar [5] reported the cystic transformation of uterine adenomyoma, and then in 1996, Tamura [6] et al. first reported a case of cystic adenomyosis in a 16-year-old woman. The incidence of cystic adenomyosis is now reported in the literature to be 5–70%. Currently, there is an increasing and younger trend, with 65–75% of women aged ≤30 years having cystic adenomyosis [7]. The main feature of cystic adenomyosis is the presence of one or more cystic cavities within the myometrium. This cystic cavity contains brownish, stale bloody fluid, and it does not communicate with the uterine cavity [8], while the cyst wall consists of endometrial glands and epithelium of the mesenchyme, surrounded by myometrial tissue. The clinical manifestations are mainly progressive and unbearable dysmenorrhea, menometrorrhagia, menostaxis, pelvic pain, infertility, etc. A few patients may be asymptomatic. A review of the relevant literature in the past 10 years showed that there were a total of 14 cases of adolescent uterine cystic adenomyosis with an average onset age of about 14 years, among which 13 cases had severe dysmenorrhea as the starting symptom. Thus, this progressive and aggravated dysmenorrhea is the main symptom of adolescent uterine cystic adenomyosis. The two patients in this study also fit this characteristic.

Juvenile cystic adenomyosis is rare, with dysmenorrhea as the primary symptom, and is highly misdiagnosed as female obstructive genital tract malformations and uterine myomatous cystic degeneration [7]. Imaging is an effective method for diagnosing the disease, especially MRI, which enables most patients to be diagnosed in early adolescence [9]. MRI not only reflects the characteristic signal of the mass and clarifies the location and size of the mass but also identifies complex uterine malformations, which is the best diagnostic modality for this disease. Its sensitivity can reach 78–88% and specificity 67–93% [10]. Because most patients are adolescents with no sexual history, the application of the transvaginal ultrasound is limited. Transabdominal ultrasound is also often misdiagnosed because of the thickness of the abdominal fat and intra-abdominal gas interference. The first patient in this study had an abdominal ultrasound suggestive of a bicornuate uterus, suggesting that cystic adenomyosis needs to be differentiated from uterine anomalies and is easily misdiagnosed as such. MRI can be used for further identification; reviewing the relevant literature from the last decade, all patients with juvenile cystic adenomyosis were easily diagnosed using MRI. Both patients in this study were also clearly diagnosed using MRI combined with clinical symptoms.

Juvenile cystic adenomyosis differs from adult cystic adenomyosis. Kriplani et al. [11] noted that adult patients with an age of onset older than 30 years, with symptoms resembling typical adenomyosis, mostly had a history of surgical trauma to the uterus. In contrast, Takeuchi et al. [8] stated the diagnostic criteria for juvenile cystic adenomyosis: (1) an age ≤ 30 years; (2) the cyst ≥ 1 cm, with the cavity independent of the uterine cavity and surrounded by hyperplastic smooth muscle tissue; and (3) an early onset of severe dysmenorrhea. However, Chun et al. [12] concluded that the older age of onset defined by Takeuchi could lead to inaccurate typing in some cases and proposed new diagnostic criteria for the adolescent type: (1) an age of onset ≤ 18 years or severe dysmenorrhea within 5 years of menarche; (2) those without a history of uterine surgical operation; and (3) those with a cystic cavity diameter ≥ 5 mm. Both patients in this study presented with severe dysmenorrhea at the age of 15 years and had no previous history of uterine operation. Therefore, they accord with the new diagnostic criteria for adolescents proposed by Chun. This criterion clearly indicates the difference between adolescent and adult forms and improves the diagnostic acuity of clinical workers in juvenile cystic adenomyosis.

Currently, there are no unified guidelines and expert consensus on the diagnosis and treatment of adolescent uterine cystic adenomyosis. Clinical treatment principles focus on symptom relief, lesion size control, and fertility protection. We analyzed the treatment of fourteen cases of adolescent uterine cystic adenomyosis in the past 10 years and found that surgery or combined drug therapy was the primary treatment. Five of the patients were unsatisfactory after pharmacological treatment. The other nine patients were mainly treated with laparoscopic cyst resection or postoperative combined drug therapy, and their symptoms were relieved after surgery. Surgical treatment is an effective way to treat uterine cystic adenomyosis. However, surgery may damage the normal muscle tissue surrounding isolated cystic adenomyosis lesions. For young women with childbearing needs, surgery may increase the risk of uterine rupture during pregnancy and may cause iatrogenic endometriosis during the procedure [13,14,15]. Therefore, surgical treatment is not well accepted by women who are nullipara. In the two patients reported in this study, the lesions were located between the myometrium, and surgical excision of the cysts was highly likely to damage the normal uterine physiological structure. Therefore, they preferred a non-invasive treatment. In a review of the literature in the past 10 years, only one patient received ethanol injection sclerotherapy, and the symptoms still existed after 5 months of follow-up. Therefore, ethanol injection sclerotherapy has poor clinical efficacy for this disease. Ryo et al. [16] reported a case of cystic adenomyosis with symptom relief after radiofrequency ablation. However, the safe and effective dose of radiofrequency ablation and ethanol is not clear, and pathological diagnosis is not available, which may lead to intrauterine adhesiveness and should be used with caution in patients with childbearing needs [17]. In recent years, the literature on the treatment of juvenile cystic adenomyosis has gradually tended toward minimally invasive or noninvasive treatment, among which HIFU has received much attention for its good safety and effectiveness in the treatment of uterine fibroids and adenomyosis [18]. It principally uses the physical characteristics of tissue penetration and the focusing and energy deposition of the ultrasound to focus in vitro ultrasound on the area of interest in the lesion to produce a thermal effect (instant temperature up to 65~100 °C), cavitation effect, and immune effect to cause irreversible changes in the lesion site, instantaneous protein degeneration, and tissue coagulation necrosis. Finally, by activating the body’s own immune mechanism, the necrotic tissue of the lesion is absorbed [19]. HIFU treatment as a new non-invasive treatment method has been reported in cases with good clinical efficacy. Zhou et al. [20] reported three cases of juvenile cystic adenomyosis with complete symptomatic remission with HIFU treatment, but with only 3–6 months of follow-up and without combined drug therapy, and whether there were subsequent fertility implications was not elaborated. Combined with the two patient’s histories, physical examinations, and auxiliary examinations, the final clinical diagnosis was juvenile cystic adenomyosis. Both patients underwent ultrasound before HIFU treatment; the ultrasonography showed a rich blood supply in the lesion. After HIFU treatment, ultrasonography showed that the lesion was reduced in size, with satisfactory grayscale changes and no blood perfusion in the lesion area. Neither of the patients had complications such as skin damage, intestinal perforation, or nerve damage during the treatment. Their dysmenorrhea symptoms were significantly relieved or even disappeared after HIFU treatment, and the lesions gradually shrank.

According to the literature, the use of GnRH-a drugs after treatment is mostly recommended to prevent recurrence [21]. The first patient reported in this study was treated with HIFU only and without combined drug therapy, and the follow-up suggested that the mass was not significantly reduced after HIFU treatment, but the dysmenorrhea symptoms were significantly relieved. The second patient was treated with HIFU that was combined with GnRH-a postoperatively, and the follow-up showed a significant reduction of the lesion, with a 97% reduction of the mass volume. Her dysmenorrhea symptoms were completely relieved and have not recurred since the follow-up. Therefore, it is worth noting that in the treatment of juvenile cystic adenomyosis, HIFU combined with drug therapy is significantly better than HIFU therapy alone for lesion size control. A study found that HIFU treatment for adenomyosis does not increase the risk of uterine rupture [22]. We reviewed the literature and found only two patients who had uterine rupture during delivery after HIFU treatment [23,24]. However, the correlation between HIFU treatment for adenomyosis and uterine rupture cannot be explained. The two patients in this study were nullipara, and close observation and follow-up were required before pregnancy, during pregnancy, and during delivery.

## 3. Conclusions

Juvenile cystic adenomyosis is a specific type of uterine adenomyosis that is rare and very easy to misdiagnose. MRI is of great value for diagnosis. Non-invasive treatment can effectively relieve symptoms, control lesions, and preserve fertility. In this report, follow-ups revealed that HIFU therapy, or HIFU combined with pharmacological therapy, can safely and effectively treat adolescent patients with cystic adenomyosis. However, there is a lack of clinical data supporting a large sample of HIFU treatments for juvenile cystic adenomyosis, and its effects on fertility remain to be further studied.

## Figures and Tables

**Figure 1 diagnostics-13-01608-f001:**
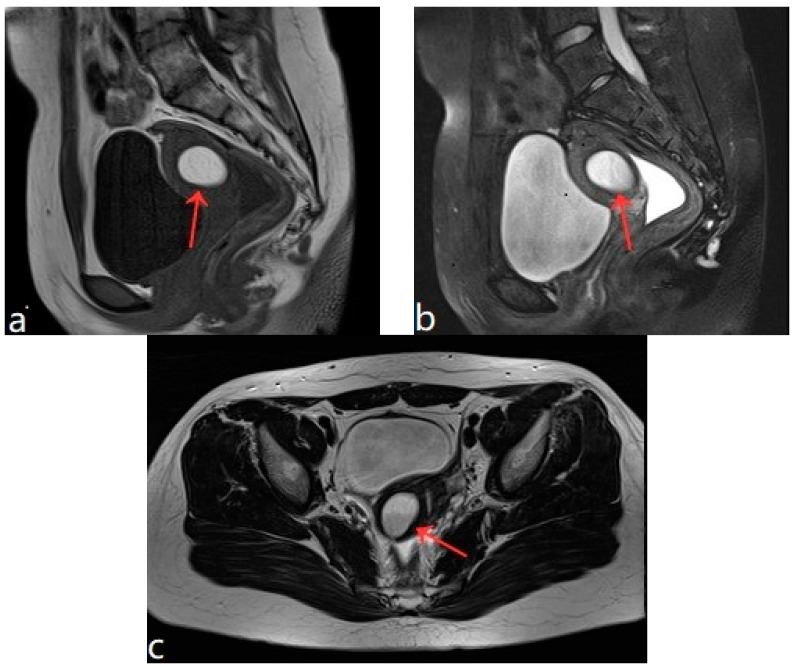
(**a**–**c**) Pelvic MRI: sagittal TIWI sequence (**a**), sagittal T2WI fat suppression sequence (**b**), and axial T2WI non-fat suppression (**c**) showing a cystic lesion in the right posterior wall of the uterus with regular morphology, oval shape, and clear border (red arrows).

**Figure 2 diagnostics-13-01608-f002:**
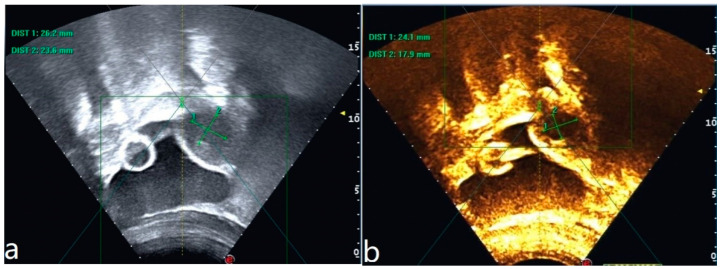
Ultrasound image of the lesion before HIFU treatment (**a**). Ultrasonography after HIFU treatment shows no perfusion of the lesion (**b**).

**Figure 3 diagnostics-13-01608-f003:**
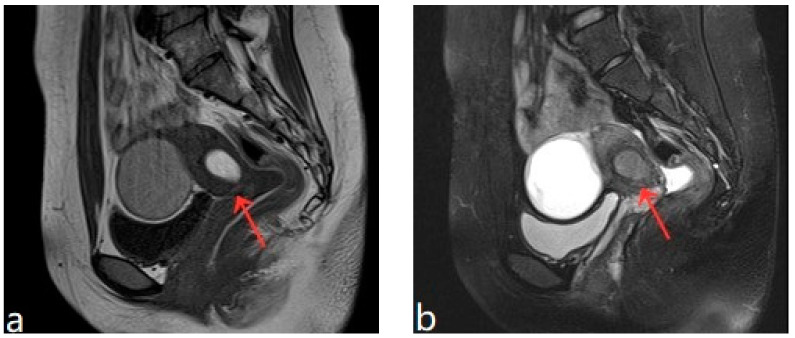
(**a**,**b**) Pelvic MRI shows lesions after HIFU treatment (1st month)(red arrows).

**Figure 4 diagnostics-13-01608-f004:**
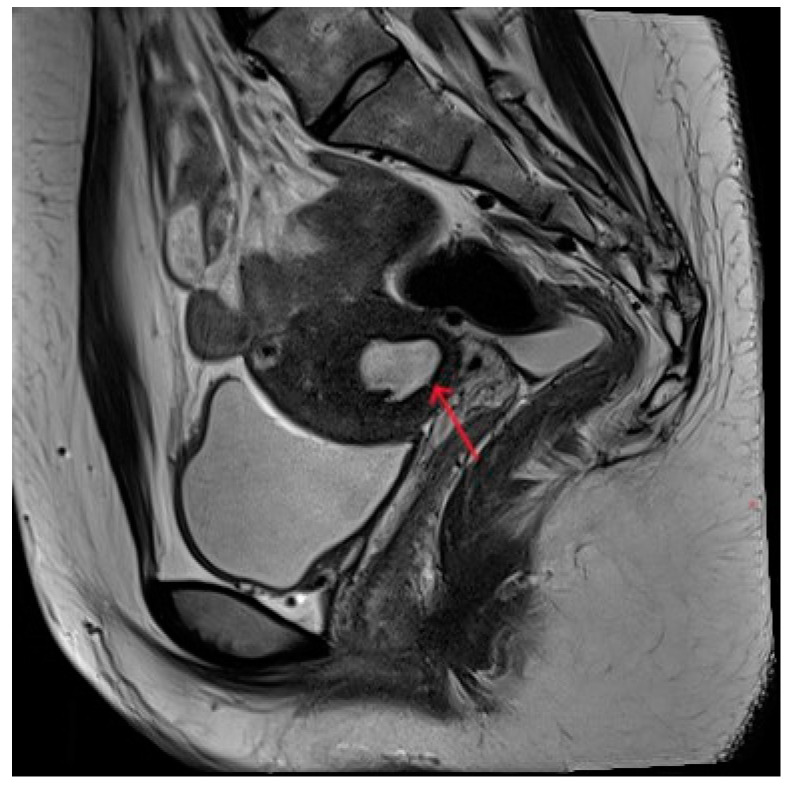
Pelvic MRI shows lesions after HIFU treatment (5th year) (red arrows).

**Figure 5 diagnostics-13-01608-f005:**
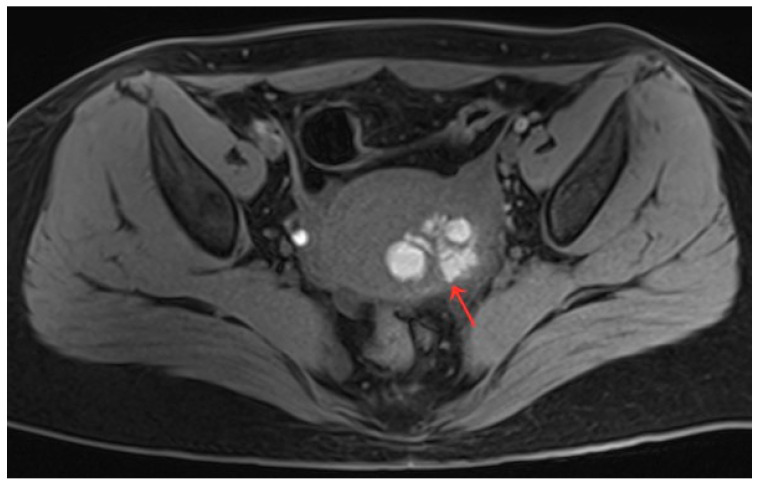
Pelvic MRI shows a cystic lesion with irregular morphology in myometrium (red arrow).

**Figure 6 diagnostics-13-01608-f006:**
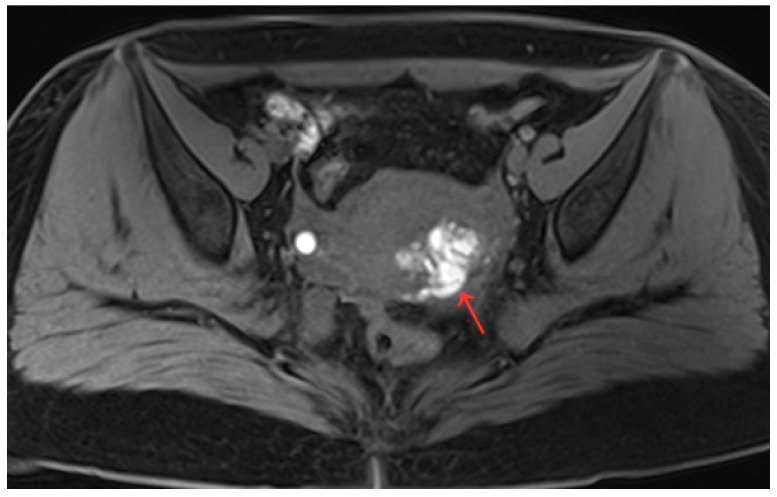
Pelvic MRI shows lesions after HIFU treatment (1st day) (red arrow).

**Figure 7 diagnostics-13-01608-f007:**
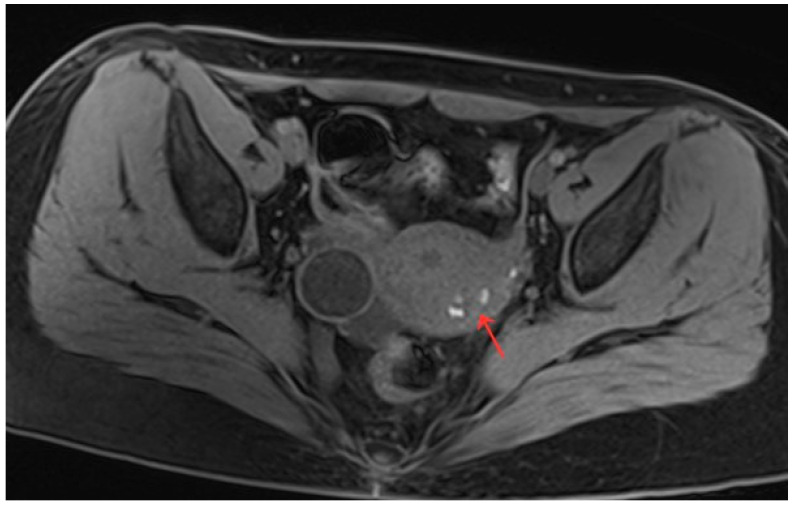
Pelvic MRI shows lesions HIFU treatment (8th month) (red arrow).

**Table 1 diagnostics-13-01608-t001:** Pre- and post-treatment parameters in two patients.

Subject	Before HIFU Treatment			After HIFU Treatment			Combined with Drugs
	Lesion Size	Symptoms	VAS Score	Lesion Size (cm)	Symptoms	VAS Score	
Case 1	2.0 cm × 3.1 cm × 2.4 cm	dysmenorrhea	7	1.1 cm × 2.4 cm × 1.0 cm ^1^	recovery	0	no
Case 2	5.1 cm × 3.3 cm × 4.7 cm	dysmenorrhea	8	1.2 cm ×1.4 cm × 1.6 cm ^2^	recovery	0	yes ^3^

^1^ Pelvic MRI shows lesions after HIFU treatment (5th year); ^2^ Pelvic MRI shows lesions after HIFU treatment (8th month); ^3^ Combined with GnRH-a injection for 4 period.

**Table 2 diagnostics-13-01608-t002:** Parameters associated with HIFU treatment in both patients.

Subject	Treatment Time	Irradiation Time	Treatment Power	Treatment Intensity	Ablation Volume Evaluation
Case 1	49 min	250 s	200 W	306/s	80%
Case 2	75 min	650 s	300 W	517/s	90%

## Data Availability

The data presented in this study are available on request from the corresponding author.

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
