# Peer review of "High Intensity Focused Ultrasound Ablation for Juvenile Cystic Adenomyosis: Two Case Reports and Literature Review"

_diagnostics, 2023, doi:10.3390/diagnostics13091608_

Round 1

Reviewer 1 Report

The manuscript presents successful HIFU treatment of juvenile cystic adenomyosis. The information is of importance and adds to the literature.
Comments:

Line17 – 25: please shorten the abstract.

Line 28: where does the information on combined drug treatment come from?

Line 34: I do not agree with the range of age. Adenomyosis is not limited to this range, but also can be found in adolescents and postmenopausal women.

Line 36: To which imaging technique do you refer? To transvaginal 2D ultrasound? Small cystic lesions (subendometrial buds or microcysts) are just one of several ultrasound signs.

Please be more precise. You can rely on the following publications:

Harmsen MJ, Van den Bosch T, de Leeuw RA, Dueholm M, Exacoustos C, Valentin L, Hehenkamp WJK, Groenman F, De Bruyn C, Rasmussen C, Lazzeri L, Jokubkiene L, Jurkovic D, Naftalin J, Tellum T, Bourne T, Timmerman D, Huirne JAF. Consensus on revised definitions of Morphological Uterus Sonographic Assessment (MUSA) features of adenomyosis: results of modified Delphi procedure. Ultrasound Obstet Gynecol. 2022 Jul;60(1):118-131. doi: 10.1002/uog.24786. PMID: 34587658; PMCID: PMC9328356.

Krentel H., et al: Accuracy of ultrasound signs in two‐dimensional transvaginal ultrasound for the prediction of adenomyosis: prospective multicenter study. 2023.
DOI: 10.1002/uog.26197
Line 39: it is important to differentiate between juvenile cystic adenomyosis and cystic myometrial lesions in older women ( as you do in the discussion section).

Line 40: what does “extremely difficult” mean? Be more precise! Differentiate between age, fertility aspects, etc.

Line 43: what does great damage mean?
Line 49: please explain the relation between the presented cases and the literature review.

Line 55: please rephrase.

Line 114: please rephrase.

Can you please provide CA12-5 levels in patient 1 after treatment?

Please add if you opted for any type of continuous hormonal treatment in both patients!

Line 160: and this is the difference in juvenile cystic adenomyosis compared to the invasive growth of adenomyosis from the cavity to the myometrium (in these cases we can find small tunnels)

Line 163: please rephrase

Line 213: what do you mean with “greatly”. I think there is no data on this question. The risk of uterine rupture is an assumption.

Line 214: what do you mean by “new medically induced endometriosis”. Please cite literature if available.

Line 218: this should be rephrased. I agree that patients should be informed in detail about the treatment options. However “they refuse surgical treatment” is too general.

Line 221: Why has RFA a poor efficacy? Where does this information come from? Please add literature.

Line 267: Please also discuss the risk of rupture after HIFU treatment.

Author Response

请参阅附件。

Reviewer 2 Report

The topic of this paper is timely and essential considering the severity of uterine cystic adenomyosis as well as the lack of a consensus with regards to the appropriate management protocol. Case description is clear and discussion section provides the appropriate evidence.

In order to improve the overall performance of the article extensive English language corrections are needed regarding grammar, syntax and phraseology.

This reviewer recommends that authors should provide a chart/figure clearly describing methodology employed during high intensity ultrasound ablation on both cases.

Moreover, a table comparing pre and post operative patient profile would be of interest assisting readership.
